# Prospect Theory: A Bibliometric and Systematic Review in the Categories of Psychology in Web of Science

**DOI:** 10.3390/healthcare10102098

**Published:** 2022-10-20

**Authors:** Júlia Gisbert-Pérez, Manuel Martí-Vilar, Francisco González-Sala

**Affiliations:** 1Departamento de Psicología Básica, Universitat de València, Avgda. Blasco Ibañez 21, 46010 Valencia, Spain; 2Departamento de Psicología Evolutiva y de la Educación, Universitat de València, Avgda. Blasco Ibañez 21, 46010 Valencia, Spain

**Keywords:** prospect theory, applied psychology, health, decision making, behavior, prevention

## Abstract

Prospect Theory (PT) is an alternative, dynamic explanation of the phenomenon of risky decision making. This research presents an overview of PT’s history in health fields, including advancements, limitations, and bibliometric data. A systematic and bibliometric review of the scientific literature included in the psychological categories of Web of Science (WoS) was performed following the PRISMA 2020 statement for systematic reviews. A total of 37 studies (10 non-empirical and 27 empirical) were included in the sample. Bibliometric results showed thematic variability and heterogeneity regarding the production, researchers, and methodologies that are used to study PT. The systematic results highlight three main fields of PT research: preventive and screening behaviors, promotion of healthy habits, and COVID-related decision making. Personal and contextual factors which alter the usual pattern specified by PT are also described. To conclude, PT currently has an interdisciplinary character suitable for health promotion, with recent studies broadening its applicability.

## 1. Introduction

Decision making under risk has been a subject of social research for several centuries. This extensive scientific interest has allowed the development of a large theoretical and experimental body on decision making under risky conditions [1], leading to new models that have attempted to solve problems such as the excessive emphasis on normativity. This paper highlights the contribution of Prospect Theory (PT).

PT was created by Kahneman and Tversky [2,3]. It developed as an alternative explanation of risky decision-making processes to Expected Utility Theory [4]. PT contemplates the presence of heuristics and limitations in human cognition, which result in biases and deviations from what is considered normative. However, these deviations are considered systematic and could be studied to improve decision making [5].

PT is based on two fundamentals. The first points out that, in deciding between the different choice options, we depend on a frame of reference and not so much on the absolute value of the options, which violates the economic conception of rationality. The second foundation of the theory is loss aversion bias. Loss aversion refers to a greater sensitivity to potential losses than to potential gains of equal magnitude [5].

To justify these assumptions, controlled experiments were developed in which participants had to choose between different alternatives (usually two) with different probabilities of achieving certain outcomes [2,6]. The obtained results showed that the decision process comprised two phases, the editing phase and the evaluation phase. First, a reference point was set and the possible outcomes were framed as benefits or losses. The process ends with a personal assessment of the usefulness of the options [2,7]. Among the basic findings and principles of Kahneman and Tversky’s theory [2,3], the S-shaped value function, the four-fold pattern of risk preferences, the “probability weighting function”, the uncertainty effect, and “the reflection effect” are worth mentioning.

PT is a descriptive theory of human behavior which does not explain how people should theoretically make their decisions, but how they actually do [8]. It has been applied, not without difficulties, to different contexts, such as economics [5,9] and politics [10,11,12]. Likewise, its assumptions have been analyzed in more specific conditions, such as energy efficiency investment [13], terrorism [14], political participation [15], or climate policies [16].

One of Kahneman and Tversky’s key insights was that the way risky decisions are framed influences what is selected, and it does so in a way captured by the assumption of an S-shaped value function defined on changes from the status quo [2,17]. Health decisions inherently involve risky choices [18]. Thus, consistent with what PT predicts, subsequent work demonstrated that the way in which health information is framed (to focus on potential gains (e.g., benefits of healthy behavior) versus losses (e.g., harms of unhealthy behavior)) systematically influences decisions and choices [17,19]. In addition, the COVID pandemic also involved risky decision making at the societal level. Consistent also with PT, gain- or loss-framing of health information influenced decision making, and risk-free behaviors may be promoted [20]. 

In addition to the framing effect, alterations in the expected pattern of loss aversion have also been studied. Regarding PT in the psychological field, its application in substance addictions stands out for its inherent risky decision making. [21]. According to PT, low levels of loss aversion increase the likelihood of engaging in addictive behaviors. Drug users have been found to show lower loss aversion than non-users [21]. All of this can be taken into account by healthcare personnel to understand the resistance and ambivalence in the decision-making processes in consumer patients.

Given its long-standing interest and applicability, the aim of this study is to conduct a bibliometric and systematic review of the PT literature in health settings within the psychology categories of Web of Science (WoS), in order to provide an overview of the usefulness, applicability and limitations of the theory within this scientific discipline. This will allow the creation of a new resource pool from which replications of previous studies, scientifically argued critiques, or even new experiments or theories can emerge, leading to more critical and informed scientific developments. It may also help psychology and health professionals to understand human cognitive issues and promote good health.

## 2. Materials and Methods

A systematic and bibliometric review of the scientific literature of Prospect Theory [2,3] in the main WoS database was conducted. A protocol was registered in PROSPERO, with identification code CRD42022348325. The search was conducted in September 2022 following PRISMA 2020 statement for systematic reviews [22]. SPSS 22 statistical package, R package Bibliometrix [23] and WoS analysis were used for the bibliometric review. 

### 2.1. Information Sources and Search Strategy

A search was performed in the Web of Science database (Core Collection) with the search term “prospect theory” and “health”. Other databases were not consulted due to the number of studies identified and the objective of exploring the WoS psychology categories.

### 2.2. Eligibility Criteria and Selection Process

In the systematic search, the inclusion criteria were (a) containing the term “prospect theory” and “health” in topic, (b) being a scientific article, (c) being included in one of the psychological WoS categories: “behavioral sciences”, “neurosciences”, “psychology”, “psychology applied”, “psychology biological”, “psychology clinical”, “psychology educational”, “psychology experimental”, “psychology mathematical”, “psychology multidisciplinary”, “developmental psychology”, “psychology psychoanalysis”, or “psychology social”, and (d) being written in English or Spanish. 

The exclusion criteria consisted o” (a)’addressing other topics (n = 80), (b) articles on other theories (n = 20), and (c) articles that were book chapters (n = 7). The selection and screening process is shown in Figure 1.

The selection process was performed by two investigators independently and then combined to reach a consensus. A third investigator supervised the results to confirm the quality of their work. 

### 2.3. Data Extraction

After the selection and analysis process, the final sample contained 37 articles. 

For the bibliometric review, the following variables were considered: year of publication, number of authors, distribution by country and continent, university affiliations, areas of research in psychology according to WoS, scientific journals, and key concepts. To perform the keyword co-occurrence networks), not all the terms were included, eliminating isolated nodes. For the systematic review, the following variables were considered: authors, year of publication, type of study, and main objective. For the empirical studies, we also extracted information on the sample, the methodology, the existence of a control group, and the main results and limitations. The bibliometric data extraction process was carried out using the WoS indicators, while data extraction for the systematic review was performed in the same way as the study selection process.

## 3. Results

### 3.1. Results of Bibliometric Review

Regarding TP production in the health field, the Figure 2 presents an irregular and increasing distribution with the highest production peak in 2021. In this year, several of the publications focused on the study and promotion of health behaviors in the COVID pandemic. The interest in applying PT to the field of health seems to have started in 1997, 18 years after the original study [2], indicating that the initial interests of this theory were focused on other fields. The last decade (2012–2022) accumulates 57% of the publications, highlighting the growing interest.

The sample includes 112 authors. Mainly, the contribution of P. Salovey (Yale University) to the field of health in PT (4 publications) stands out, followed by G. J. De Bruijn (University of Amsterdam) and A. J. Rothman (University of Minnesota System) (3 publications). The rest of the authors contribute in 2 (12% of authors) or 1 publication (86%). 

Figure 3 shows the distribution of scientific production by country, considering both internal (CMI) and international (CCM) collaborations. The sample included 12 countries in the Americas, Europe, Asia, and Oceania, and a total of 142 related publications. The USA had 81 linked publications (57%), followed by the Netherlands (16; 11%) and Canada, China, and Germany (7; 5%). Accordingly, the top five universities with the highest affiliations are Yale University (4), Maastricht University (3), University of Amsterdam (3), University of Minnesota System (3), and University of Minnesota Twin Cities (3). 

With 5 or fewer linked publications are Singapore, the UK, Australia, and France (4), Italy (3), South Korea (2), and Spain (1). Regarding international collaborations, the USA stands out with Canada and the Netherlands with two collaborations, followed by Germany–Spain–Netherlands, USA–Italy, and USA–South Korea with one collaboration.

Regarding the WoS psychology categories, the areas that appeared to be most linked to PT and health are Multidisciplinary Psychology (13 publications), Clinical Psychology (10), Psychology and Social Psychology (8), Applied Psychology (3), Experimental Psychology and Behavioral Sciences (2), and Developmental Psychology and Neurosciences (1). Among the categories that did not belong to Psychology, Economics and Public Environmental Occupational Health (2) and Gerontology, Hospitality Leisure Sport Tourism, Management, Nutrition Dietetics, Oncology, Psychiatry, Social Sciences, and Biomedical and Sport Sciences (1) stood out (Table A1).

A total of 27 scientific journals present articles related to PT and health. The scientific journals with the highest number of articles on PT in the area of health are the *British Journal of Health Psychology*, *Health Psychology*, and *Journal of Applied Social Psychology* (3, respectively). *Journal of Behavioral Medicine, Journal of Economic Psychology, Psychology Health and Social*, and *Personality Psychology Compass* have two publications each. The remaining 20 have only one publication.

By analyzing the keyword co-occurrence networks, a general picture of the predominant terms in the study of PT and health was obtained. Figure 4 shows that “prospect-theory” was the term with the highest intermediation, i.e., presenting the highest number of links to other keywords. The other terms with the least intermediation were “intentions”, “loss-framed messages”, “behavior”, “perceptions”, “information”, and “attitudes.” The size of each block indicates the frequency of occurrence as an intermediate word. The figure shows three groups of keywords (blue, red, and green) and a closer link (by the thickness of the link) between “prospect-theory-behavior” and “behavior-intentions”.

### 3.2. Results of Systematic Review

Table A1 and Table A2 in Appendix A (non-empirical studies and empirical studies, respectively) show a synthesis of the data from the studies in the sample. Ten non-empirical studies (published between 1997 and 2021) and twenty-seven empirical studies (published between 1999 and 2022) were found. 

#### 3.2.1. Prospective Theory and Health Care Field

According to DeStasio [18], there are three main contributions of PT to the health domain. First, PT indicates that people will act differently depending on whether a situation is gain- or loss-framed compared to some reference point. Second, the reference point may have a particular impact on preventive health behaviors that are unpleasant themselves (e.g., vaccinations or invasive screening tests), where the risk of the immediate negative outcome (e.g., pain) is felt higher than the risk of the potential long-term outcome. Third, PT predicts that reframing health outcomes with respect to certainty would change decisions about health behaviors (as there is often an overweighting certainty). 

PT assumes that people respond predictably to potential gains and losses. They are risk-seeking when confronted with information about losses, but risk-averse when confronted with information about gains [19]. Thus, in the health field, gain-frames may be more beneficial to promote preventive behaviors, as well as loss-frames to favor detection behaviors [24]. One possible explanation is that prevention behaviors are perceived as low risk, while detection behaviors are perceived as high risk [19,25,26]. 

There have been many examples of successful use of PT in modeling decision making in health care settings. For example, it has been used to model health behaviors such as disease treatment [26], disease prevention [27,28,29,30,31], and encouraging altruistic behaviors such as egg donation [32]. On the one hand, Fridman et al. [26] investigated the relationship between physicians’ gain-loss recommendations and prostate cancer patients’ treatment choices. Results showed that physicians’ use of loss-related words correlated with recommendations for cancer treatment, and loss words were associated with patients’ choice of treatment. On the other hand, similar results to those hypothesized by PT were obtained in disease prevention studies, but variables have been found to influence the framing effect such as cultural differences [28] and credibility of the result [30]. However, having family members with the disease to prevent did not influence decision making [31]. 

Another focus of PT study has been life attitudes in healthy and sick patients and reference point [33,34,35,36,37]. Current health status determines one’s reference point. The reference point for an advanced cancer patient with a short life expectancy will be closer to death compared to an older adult with many years of expected survival. Thus, ill patients would prefer prolonging their life over quality of life, as was found in the results [33,34,36]. Likewise, sick patients rated a mild and a severe disease situation very differently, but healthy patients rated the two scenarios as much more similar [35]. In addition, having an overly pessimistic view of old age (e.g., not correctly predicting one’s own ability to adapt to the health problems of old age) may produce a self-fulfilling prophecy, showing reduced sensitivity to loss and impacting their health behaviors (e.g., underinvesting in future health) [37].

Given PT usefulness, public health (PH) agencies could perhaps benefit from utilizing PT in a way that would optimize the effectiveness of PH messaging to increase overall local and global adherence [24]. On the one hand, expectations and disappointment regarding health may influence happiness. A practical implication would be that doctors exaggerate the risk of bad health outcomes in the future, and emphasize that patients could not have prevented bad current outcomes [38]. On the other hand, the differences in reference point in healthy and sick people can be applied to the promotion of care or insurance plans, considering the preferences of both groups [33]. Lastly, depending on the intention to prevent or treat, gain-loss frameworks can be applied to achieve attitudinal and behavioral changes [24].

Despite all the potentialities of PT, it also has weaknesses. For instance, Van’t Riet’s review [39] includes studies of framing in the health care setting with contradictory results [39,40]. Therefore, it is necessary to carry out precise analyses of the subtle differences in the messages that may influence the receptors’ reactions. 

It should be noted that in decision making, it is important to consider variables beyond framing and risk. Among the studies reviewed, personality aspects such as psychopathy, ambivalence (e.g., persistence of attitudes, resistance to change), impulsivity, anxiety, or health involvement stand out [41,42,43]. Overall, personality characteristics of the respondents played a more important role as predictors of risk choices mainly in the negative frame [42,43]. Likewise, with individuals with high ambivalence, a greater persuasion appears with a negative framing (and vice versa), due to a possible negativity bias [41].

#### 3.2.2. Prospect Theory on Promoting Healthy Habits

PT has also been used to promote health-related attitudes and behaviors, which may reduce the occurrence of diseases. In the study of the framing effects on health issues, gain frames generally had an advantage over loss frames in promoting preventive behaviors (e.g., physical activity) [44,45]. Gallagher and Updegraff [44] concluded that “how a health message is framed is an important consideration in designing messages that promote preventive behaviors’’. In this regard, a gain message was associated with better semantic and affective evaluations of the message, but also a prime/frame and frame/source valence match was found more persuasive [45]. Hence, semantic consistencies must be taken into account, as they moderate the influence of message framing.

Therefore, it makes sense that, in the case of health-affirming behaviors such as physical activity (PA), messages framed around gains (i.e., benefits) rather than losses (i.e., costs) are often more effective [19,45,46]. PT has been applied through framed messages to promote PA [47], as well as the use of fitness apps [48]. The results of these studies showed an advantage of gain-framed messages in promoting sport intentions and attitudes, self-efficacy and sport practice itself [46,48]. Likewise, the effects of the framed PA messages were studied across all age and sex groups, demonstrating that older men may especially benefit from PA messages due to a possible age-related positivity effect [47]. 

In this context, although the gain frame in PA promotion is often more effective, it is important to consider the motivations associated with PA behavior and how the frame fits with these motivations [44]. All of this implies that the effect of framed messages is not simply based on the function of detection or prevention, but that personal motivations and interpretations must be considered. In addition, a possible interaction between source credibility and frame should be considered, as the gain frame together with a credible source (e.g., a physician) indicated higher exercise intentions and behaviors [49]. 

PT has also been applied to the promotion of healthy eating. On the one hand, PT predicted that the perceived positive value (i.e., benefit) associated with accumulating gains grows in an asymptotic, rather than linear, function [2]. This function applied to healthy intake suggests that less health gain may be associated with eating more pieces of fruit, and consequently, after having eaten a piece of fruit, individuals may see less value in eating more. This hypothesis was somewhat supported; health benefits that people assign to consuming increasing amounts of fruit appear to increase, but only if consumption of a variety of fruits throughout the day is considered [50]. On the other hand, the effect of autonomy on framing effects and fruit and vegetable consumption has been studied. Churchill and Pavey [51] observed that gain-framed messages only boosted fruit and vegetable consumption among those with high levels of autonomy; therefore, autonomy moderated the framing effect.

This gain-framing effect on preventive behaviors was also present in the use of sunscreen. Individuals who read gain-framed messages compared to the loss-framed ones were more likely to ask, repeatedly apply, and use sunscreen at the beach [52]. At the neural level, these results are consistent with greater activation of the medial prefrontal cortex (MPFC) to gain-framed messages. Higher MPFC activation reliably predicts subsequent behavior [53].

Moreover, in this sense, de Bruijn [54] explored the message framing effects to promote dental health using mouth rinse for 2 weeks. Their results coincided with the promotion of preventive actions, the gain-framed information to emphasize the preventive use of mouthwash being more appropriate. No framing effects were found in the detection conditions.

Frame effect on tobacco smoking cessation has also been studied [55]. Through messages framed in gain and loss and images illustrating positive and negative consequences, it was found that the intention to quit smoking was greater when negative images (e.g., unhealthy mouths) appeared, as well as when pictures of healthy mouths illustrated the presence of preventive action. On a practical level (e.g., health campaigns), the use of fear appealing communications with vivid negative images is one way to reduce tobacco use.

#### 3.2.3. PT, COVID Pandemic, and Social Behaviors

Understanding framing effects in PH messaging is important for improving adherence, and it is particularly important when considering messaging where loss of life can be avoided, such as during the COVID-19 pandemic [24]. PT has been used to study risky decision making and the promotion of behaviors to reduce virus transmission, such as physical distancing or vaccination. People’s behavioral response to a health crisis depends on how they perceive threat and their level of risk tolerance. Through PT, public health messages can be framed to influence adherence to health recommendations, taking into account other factors that may affect adherence.

Doerfler et al. [56] focused on risky decision making during the pandemic and its relationship with Dark Triad traits. Their results coincided with those presented by Tversky and Kahneman [57]. In a gain scenario (lives saved), individuals were more likely to opt for the certain option, thereby displaying a bias toward risk-aversion. In a loss scenario (lives lost), individuals were more likely to take greater risks.

During the COVID pandemic, maintaining an adequate physical safety distance was necessary to prevent the spread of the virus, especially indoors. Neumer et al.‘s [58] online and field experiment with manipulated gain- or loss-framed messages showed that loss-framed messages were more effective than gain-framed ones promoting physical distancing. The loss-frame advantage suggests that uncertainty about the true effectiveness of distancing to avoid contracting COVID-19 is high and that people are more willing to accept this uncertainty when faced with a potential loss than gain.

Another behavior studied since PT has been vaccination during the pandemic. Vaccination is an important tool to end pandemics, but the majority of the public must be willing to be vaccinated to reach herd immunity. Using health message framing, Reinhardt and Rossman [43] conducted an online experiment with framed messages with younger and older samples. Loss frames lead to significantly more positive vaccination attitudes in younger adults than gain frames, which affects their vaccination intentions. However, the effects of gain- and loss-framed messages on vaccination attitudes and intentions in older adults did not differ significantly. This difference was explained by an age-related positivity effect in the older sample, since they ignored the negatively framed information in the loss frame condition and focused on the positive ones.

Finally, some moderators studied in relation to the framing of health message interventions during pandemics have been respondents’ age, targeted beneficiaries (self or community), uncertainty (as mentioned above), loss-framing reactance, and personality traits as psychopathy, as mentioned above [24,43,56]. In relation to health, the age of respondents may imply differences in framing effects for variables such as positivity in older people [43]. Furthermore, greater persuasion has been found when messages are directed at the respondents themselves as opposed to the general community. Reactance is directly associated with attitudes and behaviors and is expressed in negative cognitions and emotions; therefore, it may result in more negative attitudes towards the promoted behavior [56]. Lastly, psychopathy emerged as the significant predictor of risk taking during the COVID-19 crisis.

## 4. Discussion

PT is a theory that attempts to explain dynamic changes in decision making, including aspects ignored by rational choice theories and highlighting the importance of situation and value in decision making [59]. In this study, a systematic review and bibliometric analysis of the literature on PT and health-related fields included in the WoS psychology categories was performed. The results of the bibliometric analysis have shown a growing international interest in the application of PT in health issues. The USA, followed by the Netherlands and Canada, have contributed the largest amount of literature on PT in health care settings. The analysis of the co-occurrence networks showed that the most frequent terms were prospect theory, intentions, behavior, and loss-framed messages, indicating the main interests of the application of PT in health. 

Regarding the results of the systematic review, heterogeneity has been found in the topics, methodology, and even in some results. The application of PT in health has mostly focused on the framing effects to promote health behaviors and the importance of people’s reference point. On the one hand, it has generally proven useful to use a gain frame to promote preventive health behaviors, whereas a loss frame seems to be more useful for treatment or detection behaviors. Therefore, when decision making involves low risk, gain-framed messages may be more effective, as well as loss framed messages in high-risk decisions. On the other hand, current health status is a key factor in decision making, as it determines the personal reference point. Current health status can influence the choice of future treatments or preferences about longevity or quality of life. 

Other areas in relation to health that have been studied in PT have been the promotion of healthy habits. PT has been shown to be useful in promoting healthy habits, using gain-framing primarily. These behaviors, in turn, can be preventive, thus promoting wellness. Furthermore, the COVID pandemic situation has allowed numerous applications of PT in an intrinsically risky and uncertain context, especially in the promotion of preventive behaviors (e.g., social distancing, vaccination).

In summary, from a PT perspective, it is possible to encourage certain health-related behaviors depending on the framing, decision risk, and variables that may influence decision making. It is important to note that some results have been shown to be contradictory, thus requiring an analysis of the choices to be balanced, as well as consideration of variables that may influence decision making (e.g., personality traits, certainty of sources, cultural differences, age). All this can be taken into account when developing preventive or screening programs, as well as to promote healthy behaviors, considering the particularities of the targeted social sector (e.g., healthy or sick people).

In conclusion, this systematic and bibliometric review provides interdisciplinary evidence of the functionality of PT for the study of decision making under risk, highlighting both PT basis and factors that modify the expected decision patterns. Although these factors can be considered to hinder the applicability of PT, knowing its limitations can be very beneficial in extending the theory to new fronts. Understanding cognitive aspects such as decision making is essential in fields such as psychology and health, as it allows planning better assessments and interventions to promote well-being.

### 4.1. Limitations

The present study has several limitations. First, including only articles in the sample limits the complete knowledge of the study topic. Second, due to the size of the sample and the interest in the psychological categories of the WoS, other databases were not consulted. This meant that only studies categorized within the areas of psychology in WoS were included, thus providing a bibliometric and systematic approach limited to this area, which explains why studies such as the original [2] are not included in the sample. The interest in the psychological fields and PT lies in the importance of the cognitive part in decision making and its importance as a health science. Third, aspects such as the sampling method or the method of information extraction have not been considered because little information was provided in the articles in the sample.

### 4.2. Future Directions

First, a study of similar characteristics is proposed in other fields to broaden the study of PT. Second, it is proposed to conduct empirical studies that apply PT to specific fields or problems related to cognitive aspects or decision making within psychology and health, such as behavioral addictions. Third, it would be interesting to continue with the study of variables that alter the patterns expected by PT in order to extend the scientific knowledge. In this way, a more complete scientific framework would be obtained and the scope of the theory itself would be broadened. Fourth, it would be very useful to create a training program for health care and health professionals to promote preventive health behaviors and treatment. In this line, it would be interesting to test the applicability of PT with minors, in order to promote healthy habits in early ages.

## Figures and Tables

**Figure 1 healthcare-10-02098-f001:**
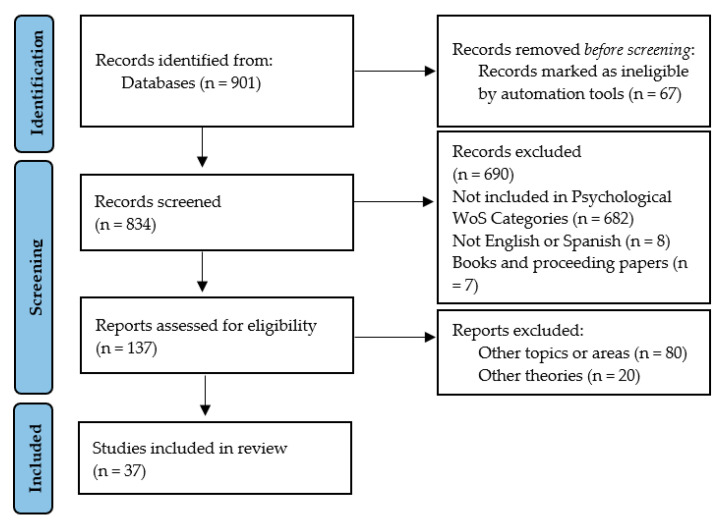
Flowchart of the selection and screening process of the systematic review articles according to the PRISMA method.

**Figure 2 healthcare-10-02098-f002:**
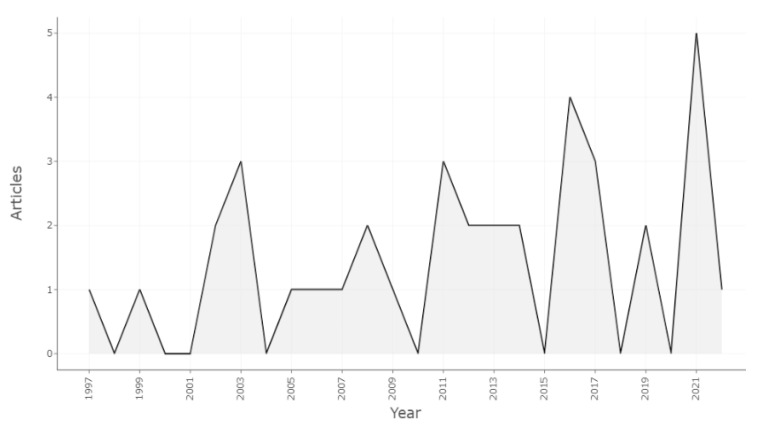
Annual scientific production.

**Figure 3 healthcare-10-02098-f003:**
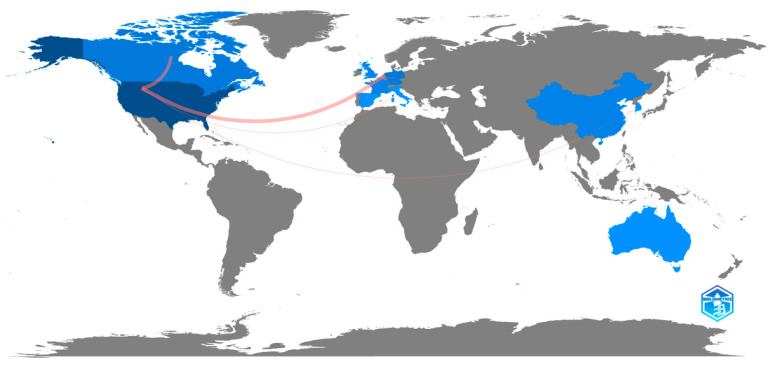
Country scientific production.

**Figure 4 healthcare-10-02098-f004:**
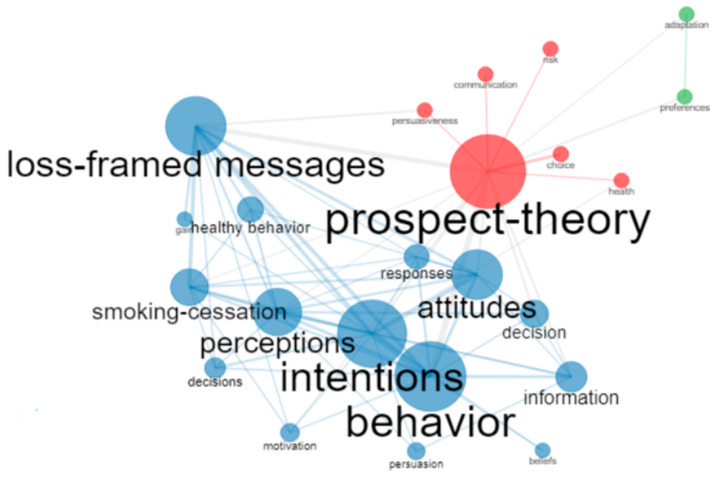
Co-occurrence network.

## Data Availability

The raw data supporting the conclusions of this article will be made available by the authors, without undue reservation.

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
