# Peer review of "Prospect Theory: A Bibliometric and Systematic Review in the Categories of Psychology in Web of Science"

_healthcare, 2022, doi:10.3390/healthcare10102098_

Round 1

Reviewer 1 Report

This paper has a number of weaknesses. First, it is either imprecisely written or the authors do not understand the topic well. Consider the following sentence from the fourth paragraph of the manuscript. 

Among the basic findings and principles of Kahneman and Tversky's theory [2,3], 38 the "V" value function, the pattern of four, the weighting function, the uncertainty effect 39 and the reflex effect are worth mentioning.

  The ""V" value function" should be "S-shaped value function", "the pattern of four" should read "the four-fold pattern of risk preferences," for a general audience "the weighting function" should be the "probability weighting function" and "the reflex effect" should read "the reflection effect."   To have this many errors about fundamental aspects of the theory you are writing about is unacceptable. There are other similar errors elsewhere in the paper (e.g., line 213, it's "construal level theory" not "construction level theory."   Second, it wasn't clear to me how you came up with the list of articles to include in your bibliometrics review. In particular, why were the 1484 articles marked ineligible by automation tools so defined? Why does the production span start in 1987 given the Kahneman and Tversky's Econometrica paper was published in 1979? It seems like the kind of information conveyed in figures 2 through 4 would be far more informative given a larger, longer sample.   Third, it is not clear to me the reviewing the findings of individual studies regarding factors that might modify predictions of PT is good way to guide practitioners as to whether it might provide a useful framework for thinking about their specific problem or area of interest. There are good articles available the attempt to synthesize findings regarding the influence of factors like framing, numeracy, etc. It might be more helpful to summarize these summaries that describe what may be the idiosyncratic findings of a handful of studies.   Finally, it is not clear to me that the topic of this paper is appropriate for a special issue aiming to  "gather research articles, reviews and/or meta-analyses on the current mental health needs of the community population."...and "review studies that address the restructuring of primary mental health care systems to better address today’s multiple challenges."

Author Response

RESPONSE TO REVIEWERS’ COMMENTS

Important changes were made to substantially improve the manuscript, and we thank the two reviewers for their necessary comments. Most of the study has been modified to bring it more in line with the theme of the special issue. We hope you find these changes appropriate, and again thank you for your reviews.

Reviewer 1

Point 1. This paper has a number of weaknesses. First, it is either imprecisely written or the authors do not understand the topic well. Consider the following sentence from the fourth paragraph of the manuscript.Among the basic findings and principles of Kahneman and Tversky's theory [2,3], 38 the "V" value function, the pattern of four, the weighting function, the uncertainty effect 39 and the reflex effect are worth mentioning.

 The ""V" value function" should be "S-shaped value function", "the pattern of four" should read "the four-fold pattern of risk preferences," for a general audience "the weighting function" should be the "probability weighting function" and "the reflex effect" should read "the reflection effect."   To have this many errors about fundamental aspects of the theory you are writing about is unacceptable. There are other similar errors elsewhere in the paper (e.g., line 213, it's "construal level theory" not "construction level theory."  

Response 1. Thank you very much for your comments and suggestions for improvement. We agree with you that the translation of some of the basic principles was not adequate. We revised the document to improve and be more precise with the terms. Therefore, those terms that were not correct have been modified.

Point 2. Second, it wasn't clear to me how you came up with the list of articles to include in your bibliometrics review. In particular, why were the 1484 articles marked ineligible by automation tools so defined? Why does the production span start in 1987 given the Kahneman and Tversky's Econometrica paper was published in 1979? It seems like the kind of information conveyed in figures 2 through 4 would be far more informative given a larger, longer sample.

Response 2. Thank you very much for your comment. Since the sample has been updated to better fit the topic of the monograph (point 4), the flow chart has also been modified (page 3). The elimination of much of the sample is due to the interest in exploring PT and its relationship to health in areas of psychology. This theory has been extensively studied in other fields, but as this was not our primary objective, we did not include a larger sample. In addition, we consider the sample of studies as starting in 1987 because the original and subsequent studies would not properly be considered from the field of psychology. However, we note the suggestion for future research, including more fields of study to obtain a more complete picture. For this reason, the following has been added to the limitations and future studies:

“Second, due to the size of the sample and the interest in the psychological categories of the WoS, other databases were not consulted. This meant that only studies categorized within the areas of Psychology in WoS were included, thus providing a bibliometric and system-atic approach limited to this area, which explains why studies such as the original [2] are not included in the sample. The interest in the psychological fields and PT lies in the importance of the cognitive part in decision making and its importance as a health science.”

“First, a study of similar characteristics is proposed in other fields of broad study of PT.”

Point 3. Third, it is not clear to me the reviewing the findings of individual studies regarding factors that might modify predictions of PT is good way to guide practitioners as to whether it might provide a useful framework for thinking about their specific problem or area of interest. There are good articles available the attempt to synthesize findings regarding the influence of factors like framing, numeracy, etc. It might be more helpful to summarize these summaries that describe what may be the idiosyncratic findings of a handful of studies.

Response 3. Thank you very much for your comment. Since we have realigned the sample to the health domain, we have also included revisions in the sample to make the results more informative. We hope it will now be more comprehensive.

Point 4. Finally, it is not clear to me that the topic of this paper is appropriate for a special issue aiming to  "gather research articles, reviews and/or meta-analyses on the current mental health needs of the community population."...and "review studies that address the restructuring of primary mental health care systems to better address today’s multiple challenges."

Response 4. Thank you very much for your constructive comment. We agree with your assessment and have therefore conducted a new search to focus on the field of health and psychology. We believe that the results now obtained may be useful for personnel in these fields and also for creating programs and training to promote health behaviors. Considering PT and cognitive aspects can be very useful in understanding behaviors and decisions and making appropriate prevention and intervention. Therefore, we have modified and added more information related to these areas throughout the paper, which can be consulted throughout the manuscript.

We hope that this more specific approach to the work will be more appropriate to the special issue.

Reviewer 2 Report

paper deals with review on Prospect theory. The work is interesting. But please address the comments for improvement

- discussion from the psychology perception must be improved in Introduction

- how it is extended to MCDM must be elaborated with graphs and tables

- other fields where Prospect Theory are popular must also be well explored with graphs and tables

- extend the review to 2022. As in Fig. 2 am able see only 2019. Expand the review for more visibility

- key works from Applied Psychology domain with respect to PT must be elaborated

- mention the limitations and future directions in detail please

Author Response

Paper deals with review on Prospect theory. The work is interesting. But please address the comments for improvement.

Thank you very much for your constructive comment. We really appreciate your words and we will revise the document to improve it.

Point 1. - discussion from the psychology perception must be improved in Introduction

Response 1. Thank you very much for your appreciation. We will include further contextualization on PT in psychology and its link to mental health. The following text has been added in the introduction, we hope it is appropriate:

“PT is particularly relevant to health decisions since they are inherently risky [17]. Previous research has shown that the way health information is framed influences individuals’ preferences and choices [18,19]. Building on PT, health messages can be framed to focus either on potential gains (benefits of healthy behavior) versus losses (detriments of unhealthy behavior). There is an ongoing debate about whether gain- or loss-framed messages more effectively promote healthy behavior, as well as which moderators produce different effects [18]. From the field of psychology, cognitive aspects such as decision making have been related to health. Its importance is evident in fields such as addictions [20], where from the perspective of PT an alteration in loss aversion has been found in consumers. Therefore, low levels of loss and risk aversion will increase the probability of showing addictive behaviors [20]. It is also a relevant theory for the study of health behaviors during the COVID pandemic, with framing being a useful aspect in promoting emotions or promoting risk-free behaviors [21].”

Point 2. - how it is extended to MCDM must be elaborated with graphs and tables

Response 2. Thank you very much for your comment. As we have modified the objective of the study to be more in line with the special issue, we are not sure if it is still necessary to include this information now. If so, please let us know where you think the information should go, as well as a little more information on what should be included.

Point 3. - other fields where Prospect Theory are popular must also be well explored with graphs and tables

Response 3. Thank you very much for your appreciation. Given that the main objective of the study is to study the contributions of PT to the field of health, we consider that this information would be more appropriate in a future study. Therefore, it has been included in the limitations:

“First, a study of similar characteristics is proposed in other fields of broad study of PT.”

Point 4.- extend the review to 2022. As in Fig. 2 am able see only 2019. Expand the review for more visibility

Response 4. Thank you very much for your comment. We have performed the search again in September 2022 to adjust it more closely to the health field. We hope it will now be more complete.

Point 5. - key works from Applied Psychology domain with respect to PT must be elaborated

Response 5. Thank you very much for your comment. We have modified the keywords to be more specific to the study. We hope they will now be more informative.

“prospect theory; applied psychology; health; decision making; behavior; prevention”

Point 6. - mention the limitations and future directions in detail please

Response 6. Thank you very much for your suggestion. In order to mention in detail the limitations and future directions, both sections have been divided and reorganized to make them clearer. We hope that it is now more detailed. Both sections now look like this:

“4.1. Limitations

The present study has several limitations. First, the inclusion of only articles in the sample limits the complete knowledge of the study topic. Second, due to the size of the sample and the interest in the psychological categories of the WoS, other databases were not consulted. This meant that only studies categorized within the areas of Psychology in WoS were included, thus providing a bibliometric and systematic approach limited to this area, which explains why studies such as the original [2] are not included in the sample. The interest in the psychological fields and PT lies in the importance of the cognitive part in decision making and its importance as a health science. Third, aspects such as the sampling method or the method of information extraction have not been considered because little information was provided in the articles in the sample.

4.2. Future directions

First, a study of similar characteristics is proposed in other fields of broad study of PT. Second, it is proposed to conduct empirical studies that apply PT to specific fields or problems related to cognitive aspects or decision making within psychology and health, such as behavioral addictions. Third, it would be interesting to continue with the study of variables that alter the patterns expected by PT in order to broaden scientific knowledge. In this way, a more complete scientific framework would be obtained and the scope of the theory itself would be broadened. Fourth, it would be very useful to create a training program for health care and health professionals to promote preventive health behaviors and treatment. In this line, it would be interesting to test the applicability of PT with minors, to promote healthy habits in early ages.”

Round 2

Reviewer 1 Report

This paper still has many weaknesses. While I appreciate that you corrected the prose describing the key features of Prospect Theory, I still very much doubt that a reader unfamiliar with the model would be able to understand why framing of choices as losses versus gains matters or why decisions involving low probabilities may differ systematically from those at higher probabilities. Your prose doesn’t help the situation. Consider the following paragraph from your paper.

PT is particularly relevant to health decisions since they are inherently risky [17]. 55 Previous research has shown that the way health information is framed influences indi- 56 viduals’ preferences and choices [18,19]. Building on PT, health messages can be framed 57 to focus either on potential gains (benefits of healthy behavior) versus losses (detriments 58 of unhealthy behavior). There is an ongoing debate about whether gain- or loss-framed 59 messages more effectively promote healthy behavior, as well as which moderators pro- 60 duce different effects [18]. 

 The logic in this paragraph is backwards. One of Kahneman and Tversky’s key insights was that the way risky decisions are framed influences what is selected and does so in a way captured by the assumption of an S-shaped value function defined on changes from the status quo. Health decisions inherently involve risky choices. Thus, consistent with what Prospect Theory would predict, subsequent work showed that the way health information is framed (to focus either on potential gains (benefits of healthy behavior) versus losses (detriments 58 of unhealthy behavior)) systematically influences decisions and choices.

The following is the next paragraph in the paper.

From the field of psychology, cognitive aspects such as decision making have been 62 related to health. Its importance is evident in fields such as addictions [20], where from 63 the perspective of PT an alteration in loss aversion has been found in consumers. There- 64 fore, low levels of loss and risk aversion will increase the probability of showing addictive 65 behaviors [20]. It is also a relevant theory for the study of health behaviors during the 66 COVID pandemic, with framing being a useful aspect in promoting emotions or promot- 67 ing risk-free behaviors [21]. 

This paragraph is just a mess. What does the first line mean? What is the “it“ the “Its” in the second line is referring to? And what does an “alteration in risk aversion” mean?

There are other instances where sentences just don’t make sense. E.g. 

According to the PT, people are more risk averse when they can gain a resource, 192 while they are more risk affine to avoid a potential loss [19]. 

What is risk affine?

These findings, while initially countering that gain frames better encourage preven- 314 tive behaviors, pandemic is a context of risk and uncertainty. 

Huh?

Author Response

Reviewer 1

Point 0. This paper still has many weaknesses. While I appreciate that you corrected the prose describing the key features of Prospect Theory, I still very much doubt that a reader unfamiliar with the model would be able to understand why framing of choices as losses versus gains matters or why decisions involving low probabilities may differ systematically from those at higher probabilities. Your prose doesn’t help the situation.

Response 0. Thank you very much for your constructive comments. We have revised all the sections to try to facilitate understanding and specify aspects that may be unclear. We have also added some sentences to complete the text of the sections that needed improvement. In addition, we have revised the writing to make it more appropriate. We hope that with the new changes it will be more understandable and complete.

Point 1. Consider the following paragraph from your paper.

PT is particularly relevant to health decisions since they are inherently risky [17]. 55 Previous research has shown that the way health information is framed influences indi- 56 viduals’ preferences and choices [18,19]. Building on PT, health messages can be framed 57 to focus either on potential gains (benefits of healthy behavior) versus losses (detriments 58 of unhealthy behavior). There is an ongoing debate about whether gain- or loss-framed 59 messages more effectively promote healthy behavior, as well as which moderators pro- 60 duce different effects [18]. 

The logic in this paragraph is backwards. One of Kahneman and Tversky’s key insights was that the way risky decisions are framed influences what is selected and does so in a way captured by the assumption of an S-shaped value function defined on changes from the status quo. Health decisions inherently involve risky choices. Thus, consistent with what Prospect Theory would predict, subsequent work showed that the way health information is framed (to focus either on potential gains (benefits of healthy behavior) versus losses (detriments 58 of unhealthy behavior)) systematically influences decisions and choices.

Response 1. Thank you very much for your constructive comment and explanation. After rereading the paragraph, we have realized that we have not expressed ourselves in a clear and understandable way, not adequately explaining some aspects as basic as the one you indicate. Thanks to the information you have provided us, we have changed this paragraph and now it is much better understood. Thank you very much for your help.

“One of Kahneman and Tversky’s key insights was that the way risky decisions are framed influences what is selected, and it does so in a way captured by the assumption of an S-shaped value function defined on changes from the status quo [2,17]. Health decisions inherently involve risky choices [18]. Thus, consistent with what PT predicts, subsequent work demonstrated that the way in which health information is framed (to focus on potential gains (e.g., benefits of healthy behavior) versus losses (e.g., harms of unhealthy behavior)) systematically influences decisions and choices [17,19].”

Point 2. The following is the next paragraph in the paper.

From the field of psychology, cognitive aspects such as decision making have been 62 related to health. Its importance is evident in fields such as addictions [20], where from 63 the perspective of PT an alteration in loss aversion has been found in consumers. There- 64 fore, low levels of loss and risk aversion will increase the probability of showing addictive 65 behaviors [20]. It is also a relevant theory for the study of health behaviors during the 66 COVID pandemic, with framing being a useful aspect in promoting emotions or promot- 67 ing risk-free behaviors [21]. 

This paragraph is just a mess. What does the first line mean? What is the “it“ the “Its” in the second line is referring to? And what does an “alteration in risk aversion” mean?

Response 2. Thank you very much for your comments. We have tried to clarify and complement what we intended to express, since it was not sufficiently clear. We hope that now it is more understandable.

“In addition, the COVID pandemic also involved risky decision making at the societal level. Consistent also with PT, gain- or loss-framing of health information influenced decision making, and risk-free behaviors may be promoted [20].

Besides the framing effect, alterations in the expected pattern of loss aversion have been studied. Regarding PT in the psychological field, its application in substance addictions stands out for its inherent risky decision making. [21]. According to PT, low levels of loss aversion increase the likelihood of engaging in addictive behaviors. Drug users have been found to show lower loss aversion than non-users [21]. All of this can be taken into account by healthcare personnel to understand the resistance and ambivalence in the decision-making processes in consumer patients"

Point 3. There are other instances where sentences just don’t make sense. E.g. 

According to the PT, people are more risk averse when they can gain a resource, 192 while they are more risk affine to avoid a potential loss [19]. 

What is risk affine?

These findings, while initially countering that gain frames better encourage preven- 314 tive behaviors, pandemic is a context of risk and uncertainty. 

Huh?

Response 3. Thank you very much for your comments. As we have commented in previous answers, we realized that some of the points we wanted to address were not sufficiently framed and explained. We hope that with the clarifications included it will be more understandable and adequate.

Regarding the first paragraph, we have modified it as follows:

PT assumes that people respond predictably to potential gains and losses. They are risk-seeking when confronted with information about losses, but risk-averse when confronted with information about gains [19].”

In the case of the second sentence, we considered that eliminating it was the best option to make the paragraph understandable. With that sentence we intended to emphasize the risk of the pandemic context itself, but we consider that it is already clear in previous paragraphs. We hope it will be more suitable now.

Reviewer 2 Report

paper can be accepted

Author Response

Reviewer 2

Point 1. paper can be accepted

Response 1. Thank you very much for your comment. Your words are greatly appreciated.
